# The Role of Zn Substitution in Improving the Electrical Properties of CuI Thin Films and Optoelectronic Performance of CuI MSM Photodetectors

**DOI:** 10.3390/ma15228145

**Published:** 2022-11-17

**Authors:** Chien-Yie Tsay, Yun-Chi Chen, Hsuan-Meng Tsai, Phongsaphak Sittimart, Tsuyoshi Yoshitake

**Affiliations:** 1Department of Materials Science and Engineering, Feng Chia University, Taichung 40724, Taiwan; 2Department of Advanced Energy Science and Engineering, Kyushu University, Fukuoka 816-8580, Japan

**Keywords:** metal halide semiconductor, cuprous iodide, zinc substitution, spin-coating, MSM photodetector

## Abstract

Pure CuI and Zn-substituted CuI (CuI:Zn) semiconductor thin films, and metal-semiconductor-metal (MSM) photodetectors were fabricated on glass substrates by a low-temperature solution process. The influence of Zn substitution concentration (0–12 at%) on the microstructural, optical, and electrical characteristics of CuI thin films and its role in improving the optoelectronic performance of CuI MSM photodetectors were investigated in this study. Incorporation of Zn cation dopant into CuI thin films improved the crystallinity and increased the average crystalline size. XPS analysis revealed that the oxidation state of Cu ions in all the CuI-based thin films was +1, and the estimated values of [Cu]/[I] for the CuI:Zn thin films were lower than 0.9. It was found that the native p-type conductivity of polycrystalline CuI thin film was converted to n-type conductivity after the incorporation of Zn ions into CuI nanocrystals, and the electrical resistivity decreased with increases in Zn concentration. A time-resolved photocurrent study indicated that the improvements in the optoelectronic performance of CuI MSM photodetectors were obtained through the substitution of Zn ions, which provided operational stability to the two-terminal optoelectronic device. The 8 at% Zn-substituted CuI photodetectors exhibited the highest response current, responsivity, and EQE, as well as moderate specific detectivity.

## 1. Introduction

Cuprous iodide (CuI) consists of nontoxic elements, and it is one of the most promising p-type wide-bandgap (~3.1 eV) metal halide semiconductor materials. It has attracted great research interest because of its widely adjustable electrical conductivity, which can be varied from a conducting electrode to a semiconducting active layer in electronic and optoelectronic devices through modulation of the process parameters and chemical composition (i.e., by doping or alloying approaches) [1,2]. Furthermore, it can be used to prepare device-quality thin films via various physical vapor deposition techniques and chemical solution methods at a low processing temperature (<200 °C). These techniques include reactive sputtering, pulsed laser deposition, thermal evaporation, vapor or solid iodization, chemical bath deposition, spray coating, and spin-coating [3,4,5]. These advantages allow flexible integration with the fabrication processes of solid-state microelectronic components.

CuI-based thin films have been used for photovoltaic cells, heterojunction diodes, field-effect transistors, thermoelectric devices, and photodetectors due to their superior electrical properties [6,7,8]. The Cu 3d orbital plays a critical role in the electrical properties of CuI semiconductors, and the native defect copper vacancy (V_Cu_) has low formation energy and allows easy creation of shallow acceptor states above the valence band maximum (VBM) [1,9]. Due to that feature, γ-CuI thin films possess p-type conductivity, as well as high conductivity with a high hole concentration (>10^17^ cm^−3^) [1].

Zhu et al. investigated the physical properties of CuI doped with group II and III elements using density functional theory, and they commented that the solubility of the cation dopants is primarily determined by the difference in electronic configurations between dopant and host [10]. It is agreed that the stable oxidization state of Cu ions in CuI crystals is +1 [1]. The electronic configurations of Zn^2+^ (3d^10^4s^2^) and Cu^+^ (3d^10^4s^1^) are similar [10]. They have the same tetrahedral coordination structure, and the ionic radius of Zn^2+^ (74 pm) is slightly smaller than that of the Cu^+^ (77 pm) [11,12]. Therefore, Zn^2+^ ions can easily be substituted for Cu^+^ sites in CuI crystals without serious distortion of the host lattice [7]. In addition, Zn^2+^ has a higher valence state than that of Cu^+^, so it can be utilized as an impurity dopant to reduce the excessive hole concentration of pristine CuI thin films. Moreover, the formation energy of the [Zn_Cu_ + V_Cu_] complex is lower than those of other defect complexes of cation dopants [10]. Xia and colleagues have reported that Zn^2+^ ions can occupy the Cu sites in CuI lattices and serve as donors, allowing the production of n-type CuI-based semiconductor thin films [11].

A photodetector is a two-terminal thin film optoelectronic device for converting optical energy into an electrical signal [13]. Photodetectors have been widely utilized in environmental monitoring, industrial production, military defense, and space research; they also hold promise for use in smart cities and autonomous vehicles [14,15]. The p-type CuI semiconductor thin films have been integrated with various n-type semiconductor materials, such as silicon (Si), zinc oxide (ZnO), and indium gallium zinc oxide (IGZO), to construct p-n heterojunction ultraviolet (UV) photodetectors to improve their optoelectronic performance or to realize self-powered photodetectors [6,8]. The structure of a metal−semiconductor−metal (MSM) photodetector is the simplest and easiest to fabricate of all semiconductor photodetectors [13]. The preparation of MSM photodetectors is a great approach for investigating the electrical and optoelectronic characteristics of semiconductor materials. For example, Huang et al. fabricated flexible CuI MSM photodetectors on PI substrates by the vacuum evaporation method, and these photodetectors exhibited good optoelectrical properties and bending stability under 1 V bias [6].

The solution-based coating approach is a cost-effective manufacturing method for the deposition of optoelectronic thin films because it does not require expensive high-vacuum chambers or other costly equipment. Additionally, the solution process is simple and easy to perform, and it allows convenient and efficient incorporation of foreign atoms for flexible adjustment of the chemical composition so as to modulate the optical and electrical properties [16]. In this study, CuI-based semiconductor thin films were deposited on glass substrates by the spin-coating technique and then annealed at a low temperature of 150 °C. The authors systematically investigated the influence of Zn substitution on the microstructural, optical, and electrical properties of solution-processed CuI thin films by varying the Zn concentration from 0 to 12 at%. In addition, these as-prepared device-quality CuI-based thin films were utilized as the photosensing layers in MMS structured photodetectors for studies of their optoelectronic characteristics.

## 2. Materials and Methods

Pure cuprous iodide (CuI) and Zn-substituted CuI (CuI:Zn) thin films were deposited on Corning EAGLE XG glass (Corning Inc., Corning, NY, USA) substrates by the spin-coating technique and subsequent thermal annealing. Each glass substrate (dimensions: 50 × 50 × 0.7 mm^3^) was ultrasonically cleaned with acetone and ethanol for 10 min, respectively, and then dried on a hot plate prior to the deposition of metal halide thin films. The 0.5 M precursor solutions were prepared by mixing CuI powder (Strem Chemicals), ZnI_2_ powder (Alfa Aesar), 2-methoxyethanol (2-ME, Acros Organics), and monoethanolamine (MEA, Acros Organics), after which they were stirred at 60 °C for 3 h to ensure complete dissolution. The Zn substitution level ([Zn]/[Cu] + [Zn]) was varied from 0 to 12 at% in increments of 4 at%, and the molar ratio of 2-ME to MEA was 1:1. The spin-coating procedure was performed at a rotation speed of 3000 rpm for 30 s. Each spin-coated film was annealed in air at 150 °C for 1 h to form CuI phase thin film and improve the densification and crystallinity. In addition, Ni interdigital electrodes were deposited onto the CuI-based semiconductor thin films with a fine metal shadow mask via vacuum thermal evaporation for fabrication of metal−semiconductor−metal (MSM) structured photodetectors. The thickness of the interdigital electrodes was about 50 nm; the electrode spacing, width, and length were 165 μm, 90 μm, and 3 mm, respectively. A schematic device structure of the MSM photodetector is illustrated in Figure 1.

The phases and crystal structures of the as-deposited CuI-based thin films were identified with a Bruker grazing incidence X-ray diffraction (GIXRD) diffractometer (Bruker, Billerica, MA, USA). The cross-sectional and plane views of the film microstructures were observed with a Hitachi field-emission scanning electron microscope (FE-SEM, Hitachi High-Technology, Tokyo, Japan). For X-ray photoelectron spectroscopy (XPS), an ULVAC-PHI PHI 5000 VersaProbe (ULVAC-PHI, Chigasaki, Japan) was used to analyze the chemical composition and examine the electronic states. Photoluminescence (PL) emission spectra were recorded with a HORIBA Jobin-Yvon LabRAM HR Micro-PL spectrometer (HORIBA Jobin Yvon, Paris, France) with a 325-nm He-Cd laser as the excitation source. A Hitachi U-2900 double beam ultraviolet-visible (UV-Vis) spectrophotometer (Hitachi High-Technology, Tokyo, Japan) was used to measure the light transmission property to determine the optical characteristics. The electrical characteristics were measured with an Ecopia HMS-3000 Hall measurement system (Ecopia, Anyang, Republic of Korea) with a magnetic field of 0.55 T and the van der Pauw configuration. Current−voltage (I–V) and current−time (I–t) measurements were performed with a Jiehan 5600 source-measure unit (Jiehan, Taichung, Taiwan) in a dark environment and under UVA light illumination. Ultra violet tubes (Philips lighting, Eindhoven, The Netherlands) provided incident power density of 2.1 mW/cm^2^ for measurement of the transient photoresponse. All the measurements of the physical properties were performed at room temperature.

## 3. Results and Discussion

### 3.1. Physical Properties of Solution-Processed CuI-Based Thin Films

The crystal structures of the spin-coated pure CuI and Zn-substituted CuI (CuI:Zn) thin films deposited on glass substrates and thermally annealed were identified with recorded X-ray diffraction (XRD) patterns, as shown in Figure 2a. XRD results indicated that the as-prepared CuI-based thin films were all polycrystalline phase, and three observable diffraction peaks and a weak diffraction signal corresponded to the (111), (220), (311), and (200) crystallographic planes of International Centre for Diffraction Data (ICCD) card number 06-0246 for a zinc-blende structure of γ phase CuI crystals. These thin films still maintained the original zinc blende lattice structure, in spite of the incorporation of Zn ions into CuI crystals to fill Cu vacancies and/or replace Cu ion sites. In addition, no diffraction signals of CuO or Cu_2_O were detected. When the Zn substitution concentration was further increased to 8 at%, the compressive stress resulting from the substitution of Zn^2+^ ions caused a slight shift of the (111) diffraction peak toward the low diffraction angle region.

These XRD patterns also revealed a shoulder peak on the low angle of the diffraction 2θ side, centered at around 2θ = 24.5° (Figure 2b) and corresponding to precipitation of the I phase. It was found that the relative diffraction intensity of (I_I(111)_/I_CuI(111) + I(111)_) decreased from 0.250 to 0.096 when the Zn substitution level was increased from 0 to 12 at% (the first column of Table 1), indicating that the impurity dopant increased the stability of the CuI phase. The average crystallite sizes of the four metal halide thin films were calculated using Scherrer’s equation [17] from the X-ray wavelength, full-widths at half-maximum, and diffraction angles of the three major diffraction peaks, including (111), (220), and (311). The estimated results, listed in Table 1, revealed that the pure CuI thin film had an average crystallite size of 7.23 nm, while that of CuI:Zn thin films increased with higher Zn substitution concentrations. When the polycrystalline CuI thin films were doped with Zn, the Zn^2+^ ions mainly filled the Cu vacancies, leading to decreases in the concentration of Cu vacancies and improvements in the crystallinity of CuI-based thin films.

The thicknesses of the CuI-based thin films were determined by cross-sectional FE-SEM observation. Typical FE-SEM micrographs of fractured cross-sections of the pure CuI and 8 at% Zn substituted CuI thin films are presented in Appendix A. The thicknesses of the CuI-based thin films were estimated to be 285–295 nm. From the plane-view FE-SEM micrographs, shown in Appendix A, it was clearly observed that the 8 at% Zn substitution sample had a larger grain size (26.5 nm) than that of the pure CuI sample (11.2 nm); that feature was in good agreement with the XRD results.

Wide-scan XPS surface analysis confirmed the presence of the constituent elements of Cu, I, and Zn in the as-prepared CuI-based thin films, (Appendix A) and XPS fine-scan analysis revealed that the intensity of Zn 2p increased almost linearly with increases in Zn substitution concentration, indicating the successful incorporation of the expected amounts of the impurity dopant Zn into the CuI crystals (data not shown). The examined chemical compositional ratio (I/Cu) for the pure CuI thin film was slightly deviated from the expected ratio of 1 and similar to a reported result for solution-processed CuI thin films [18]. Fine-scan XPS examination also identified the chemical bonding states of Cu and I in these obtained thin film samples. Figure 3a,b shows the core level XPS spectra of Cu 2p and I 3d, respectively. The Cu 2p and I 3d orbitals both split into two peaks due to the spin-orbit interaction, which was consistent with the reported XPS fine-scan spectra of CuI thin films prepared by vacuum thermal evaporation [3]. In the Cu 2p spectra shown in Figure 3a, the binding energies of the peaks located at 951.4 eV and 931.6 eV corresponded to the Cu 2p_1/2_ and Cu 2p_3/2_ states, respectively [19]. The non-existent satellite peaks indicated that no +2 oxidation state copper ions, Cu^2+^, formed, and this absence also implied the presence of only +1 oxidation state copper ions, Cu^+^ [20,21]. In addition, it was observed that the spectra of Cu 2p_3/2_ state exhibited a small asymmetry feature on the lower binding energy side, with the sub-spectra centered at about 930 eV, which is the origin of I 3p_1/2_ state [22]. As shown in Figure 3b, the peaks positioned at 630.4 eV and 618.8 eV were consistent with the I 3d_3/2_ and I 3d_5/2_ states [19]. The differences between the two split peaks of Cu 2p and I 3d spectra for the pure CuI thin film were 19.8 eV and 11.6 eV, which were similar to the values reported for transparent CuI thin films obtained by a solution/vapor two-step strategy [21]. It was noted that the spectra of Cu 2p and I 3d shifted slightly toward the low binding energy region at higher concentrations of Zn, indicating that binding energy was reduced slightly with increases in Zn substitution concentration.

The atomic ratio of copper to iodine, [Cu]/[I], was calculated using the relative sensitivity factor method [3]. The formula is expressed as
(1)$$\frac{\left[\mathrm{Cu}\right]}{\left[I\right]}=\frac{I_{\mathrm{Cu}}/S_{\mathrm{Cu}}}{I_{I}/S_{I}}$$
where I_Cu_ and I_I_ are the peak intensities of the Cu 2p_3/2_ and I 3d_5/2_ peaks, and S_Cu_ and S_I_ are the atomic sensitivity factors of Cu and I, obtained from XPS examination as 2.626 and 6.302, respectively. The estimated [Cu]/[I] values were 0.84 to 0.90 for the three CuI:Zn thin film samples, indicating that they possessed excessive amounts of I^−1^ ions [3].

The PL spectra of the four CuI-based thin films were deconvoluted into two Gaussian curves, and the two violet emission peaks were denoted as I and II, respectively (Figure 4). The results revealed that all the thin film samples exhibited 411.7 nm and 426.5 nm emissions under 325 nm-laser excitation. The strong emission peak located at 411.7 nm originated from the near-band-edge transition due to the band-to-band recombination [23]; the emission peak centered at 426.5 nm was attributed to the recombination of the donor−acceptor pair [11]. It is noted that the intensity of the 411.7 nm emission (peak I) was significantly improved with increases in the Zn substitution concentration. At the same time, the full-width at half-maximum of the peak I emission decreased with increases in Zn concentration. These features indicated that Zn substitution could enhance the crystallinity of CuI thin films. We also found that the relative emission intensity of peak I to peak II rose from 0.32 to 0.37 when the Zn concentration in the coating solution was increased from 0 to 12 at%. Chen et al. suggested that the photoluminescence at 428 nm to 431 nm was associated with Cu vacancies in CuI lattices [23], and Sirimanne et al. reported that this feature was closely related to the conductivity of CuI thin films [24]. The Zn incorporated into CuI could serve as donors and reduce the concentration of acceptors.

The absorption coefficient (α) at a specific wavelength was calculated as α = [ln(1/T)]/d, where d is the film thickness and T is the transmission coefficient [25]. Absorption spectra of the Zn-substituted CuI thin films with Zn substitution concentrations of 0–12 at% deposited on glass substrates in the wavelength range of 340–610 nm are depicted in Figure 5a. Each absorption spectrum exhibited sharp drops at 405 nm to 410 nm, corresponding to excitonic band edge absorption and exhibiting a near UV response nature.

It is well known that the optical bandgap energy (Eg) of direct allowed transitions can be estimated from the relationship of (αhν)^2^ ∝ (hν − Eg), where hν is the photon energy [25]. Figure 5b shows the curves of absorption energy (αhν) versus photon energy (hν), namely, the Tauc plots, of the four CuI-based thin films. The optical bandgap energy was determined from the Tauc plot by extrapolating a tangential line from the region of the absorption edge to the photon energy axis. The results revealed that all the CuI-based thin films had wide bandgaps approaching 3.0 eV; those values were the same as the results for spin-coated Zn-doped CuI thin films annealed at 80 °C reported by Liu et al. [12] and CuI thin films prepared via vapor iodization of copper films reported by Lin et al. [26].

The Urbach energy (E_U_) is used for studying the structural disorder and concentration of defect states of semiconductor thin films [27]. It can be estimated from the formula of α = α_0_ exp (hv/E_U_), where α_0_ is a constant. Figure 5c shows the plot of variation of ln (α) with photon energy (hν). The Urbach energy was determined by calculating the inverse of the slope in the linear region of the absorption curve [28], as presented in Table 1. The estimated results showed that the E _U_ value of the pure CuI thin film was higher than that of the CuI:Zn thin films, and when the Zn concentration was increased from 4 to 12 at%, it decreased from 77.1 meV to 72.8 meV. This change was attributed to the reduced density of defect states in the forbidden band, which improved the quality of the CuI-based thin films.

The Zn^2+^ ions incorporated into the CuI crystals filled the Cu vacancies and/or replaced some of the Cu^+^ sites, and they also provided free electrons. The measured electrical properties, including major carrier concentration (hole or electron), Hall mobility (μ), and electrical resistivity (ρ), of the obtained CuI-based thin films are summarized and listed in Table 1. Hall-effect measurement confirmed that the pure CuI thin film had p-type conductivity, and all the Zn-substituted CuI thin films were converted to n-type conduction. The carrier concentration and electrical resistivity of the pure CuI thin film were compared with those in a previous report of polycrystalline CuI thin films prepared by spin-coating and annealing in air (p = 2.5 × 10^18^ cm^−3^, ρ = 5.0 × 10^−1^ Ω cm) [5]. The carrier concentration of CuI:Zn thin films was in the range of −4.0 × 10^18^ to −4.45 × 10^18^ cm^−3^, and the electrical resistivity tended to decrease as the Zn substitution concentration increased. The electrical properties of the three CuI:Zn thin film samples were close to those of the 4 mol% Zn-doped CuI thin film fabricated by the vacuum evaporation technique (n = 5.08 × 10^18^ cm^−3^, μ = 9.68 cm^2^/Vs, ρ = 1.27 × 10^−1^ Ω cm) [11]. The Hall mobility is the combination of the mobilities of impurity scattering and grain boundary scattering. The measured Hall mobilities of the CuI:Zn samples were almost three times higher than that of the pure CuI sample, and the mobility slightly increased with increases in Zn concentration from 4 at% to 12 at%. These results can be ascribed to the enhancement of crystallization and increases in grain size, which in turn led to the reduction of grain boundary scattering. The exploration of n-type CuI-based semiconductor thin films is important for fabricating switching semiconductor devices and will contribute to the development of CuI p-n homojunction optoelectronic devices.

### 3.2. Optoelectronic Properties of the CuI-Based MSM Photodetectors

The electrical properties of the semiconductor sensing layer greatly affect the optoelectronic performance of a photodetector. The above discussion demonstrated that the electrical properties of solution-processed CuI semiconductor thin films could be tuned by controlling the amount of Zn^2+^ addition. Figure 6a,b compare the current−voltage (I–V) characteristic curves of four CuI-based MSM photodetectors under applied bias voltage across the two interdigitated electrodes and swept from −5 V to +5 V both in a dark environment and under the illumination of UVA light. It is noted that the pure CuI device exhibited the lowest dark current, and the recorded dark current of the CuI:Zn devices displayed a trend of increases with higher Zn substitution levels at the same bias voltage. The inverse of the slope of the I–V characteristic curve corresponded to the resistance of the as-fabricated two-terminal device, based on Ohm’s law (R = V/I). The resistances of the 0, 4, 8, and 12% Zn-substituted CuI MSM photodetectors in the bias voltage range of 0 V to 1 V were 1.7 × 10^4^, 1.3 × 10^4^, 1.0 × 10^3^, and 6.7 × 10^2^ Ω, respectively. That tendency is in good agreement with the electrical resistivity of CuI-based thin films with varied Zn substitution concentrations (the last column of Table 1). In addition, the slope of the I–V characteristic curve of each corresponding device was higher in the illuminated state than in the dark environment. The reason was that the CuI-based semiconductor layer absorbed the energy of the UVA light, which led to the creation of electron-hole pairs and photo-generated carriers, which formed a photocurrent when an external bias voltage was applied.

Figure 7 shows the time-resolved photoresponse characteristics of CuI-based MSM photodetectors under illumination by UVA light with an intensity of 2.1 mW/cm^2^ at 0.5 V bias voltage. The measured results of dynamic photoresponse displayed three consecutive on/off switching cycles. When the UVA light was turned on, the photoresponse currents increased with exposure time and achieved an approximate maximum level; when the UVA light was turned off, the recorded currents significantly decreased with operating time and exponentially decayed, returning to the original current level. The CuI:Zn MSM photodetectors exhibited a stable operation with a high degree of consistency, indicating that these developed devices had good photo-switching behavior, operational stability, and reproducibility. The optoelectronic performance of two-terminal CuI-based photodetectors was evaluated in terms of several important photodetection properties, including response current (I_light_-I_dark_), sensitivity (S), responsivity (R), specific detectivity (D), and external quantum efficiency (EQE) [8,18]. The calculated mathematic relation, parameter definitions, and calculated results for important photodetection properties are summarized in Table 2. As shown in that table, the CuI:Zn photodetectors exhibited better photoresponse characteristics than those of the pure CuI photodetector, including improved response current, responsivity, and external quantum efficiency (EQE). The reason was that the Zn incorporated into the CuI semiconductor thin films enhanced the film’s quality and improved the electrical properties, especially the carrier mobility and electrical resistivity. It was noted that photodetection properties of the 12 at% Zn substituted CuI photodetectors were degraded more than those of the other two CuI:Zn photodetectors. Although the semiconductor sensing layer of the former had the lowest electrical resistivity among the CuI-based thin films, it led to a significant rise in the dark state current and caused degradation of the photoresponse characteristics. In this study, the 8 at% Zn substituted CuI photodetectors exhibited the highest response current of 2.05 × 10^−4^ A, the best responsivity of 722 mA/W, EQE of 242%, and good specific detectivity of 1.51 × 10^8^ J.

Table 3 presents the structural features and important photoelectronic properties of the solution-processed ZnO- and CuI-based MSM photodetectors of the present study for comparison with those of previous studies. It was found that the processing temperature of CuI-based devices was much lower than that of the ZnO-based devices, and also that the photoresponse properties of the CuI-based devices were better than those of the ZnO-based devices. These differences can be ascribed to higher magnitude major carrier concentration of the former than of the latter, even though the latter possessed a low dark current. This study realized the development of MSM photodetectors with great optoelectronic performance based on low-temperature solution-processed CuI-based semiconductor thin films and suggested a potential metal halide semiconductor for the fabrication of flexible MSM photodetectors.

## 4. Conclusions

CuI-based semiconductor thin films and MSM UV photodetectors have been fabricated on glass substrates by the spin-coating technique at a low-temperature of 150 °C. The as-prepared polycrystalline CuI-based thin films had a wide optical bandgap approaching 3.0 eV, and the substitution of Zn^2+^ ions into CuI crystals decreased the Urbach energy and enhanced the crystallinity. XPS analysis indicated that the oxidation state of Cu ions was +1, and the CuI:Zn thin film samples possessed excessive amounts of I ions. Hall measurement results revealed that the CuI semiconductor thin films were converted from p-type to n-type conductivity after the incorporation of Zn^2+^ ions into CuI nanocrystals, and the incorporated Zn^2+^ improved the electrical properties, including the electrical resistivity and Hall mobility. The CuI:Zn photodetectors had obviously enhanced photoresponse characteristics as compared with those of the pure CuI photodetector under UVA light illumination. In the present study, the 8 at% Zn substituted CuI photodetectors exhibited the highest response current of 2.05 × 10^−4^ A, the best responsivity of 722 mA/W, EQE of 242%, and good specific detectivity of 1.51 × 10^8^ J.

## Figures and Tables

**Figure 1 materials-15-08145-f001:**
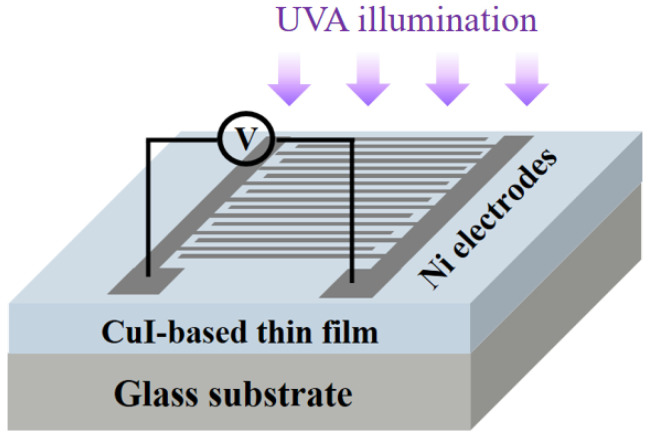
Schematic diagram of CuI-based MSM photodetector structure.

**Figure 2 materials-15-08145-f002:**
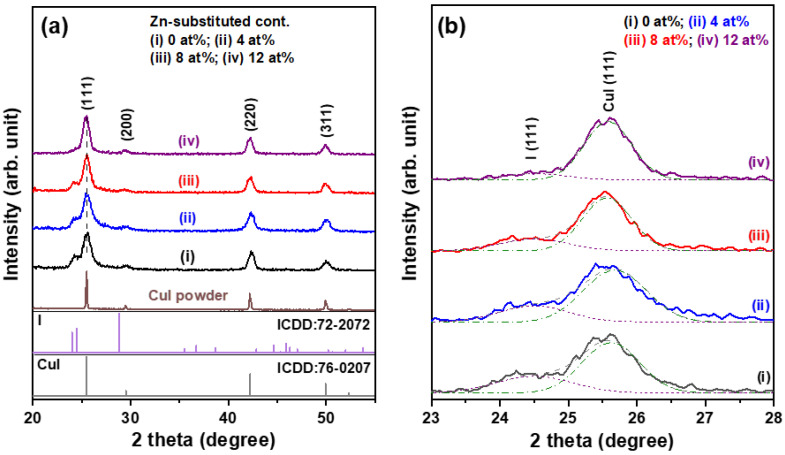
(**a**) XRD patterns of solution-processed Zn-substituted CuI thin films. (**b**) an enlargement plot of the CuI (111) diffraction peaks for the corresponding thin film samples.

**Figure 3 materials-15-08145-f003:**
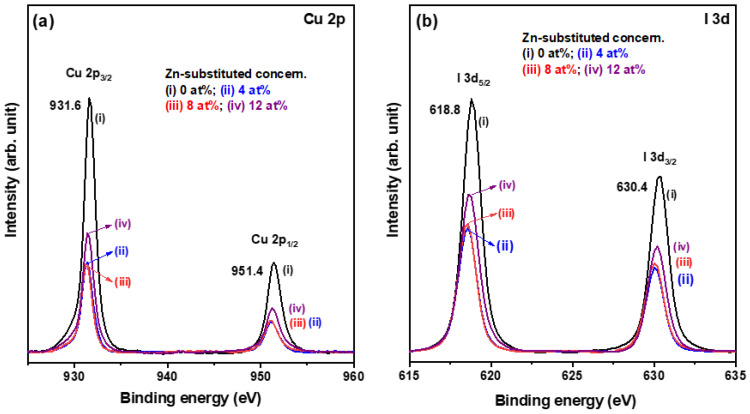
XPS spectra of the core level regions of (**a**) Cu 2p and (**b**) I 3d for the Zn-substituted CuI thin films.

**Figure 4 materials-15-08145-f004:**
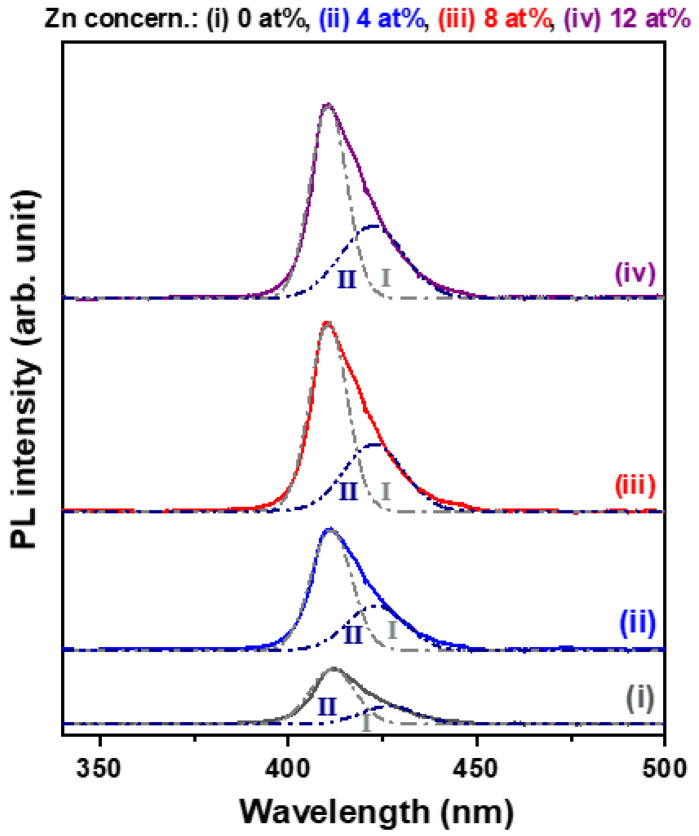
Room-temperature PL spectra of the Zn-substituted CuI thin films.

**Figure 5 materials-15-08145-f005:**
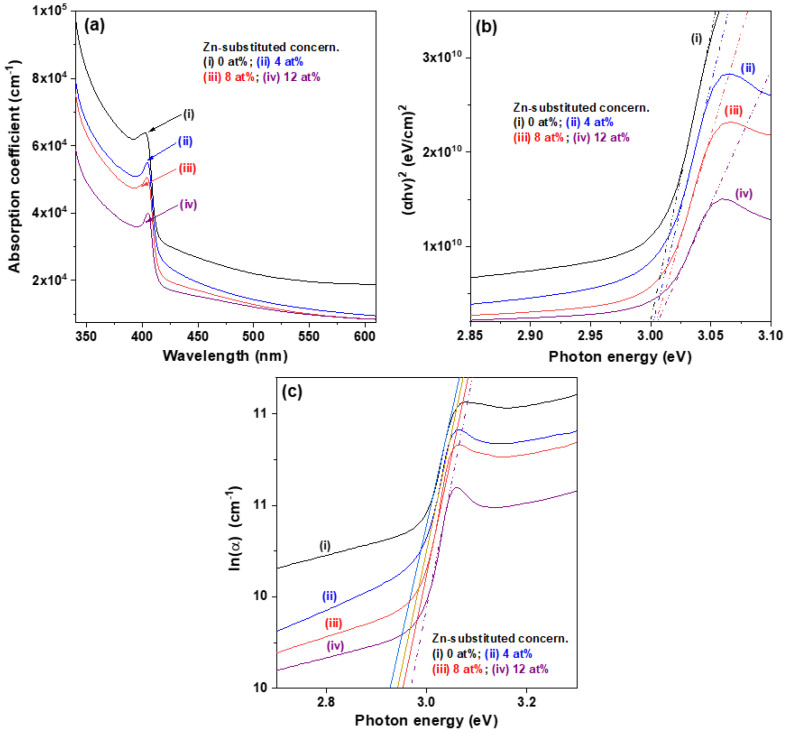
(**a**) UV-visible absorption spectra, (**b**) plot of (αhν)^2^ versus photon energy (hν), and (**c**) plot of ln (α) versus photon energy (hν) for the Zn-substituted CuI thin films.

**Figure 6 materials-15-08145-f006:**
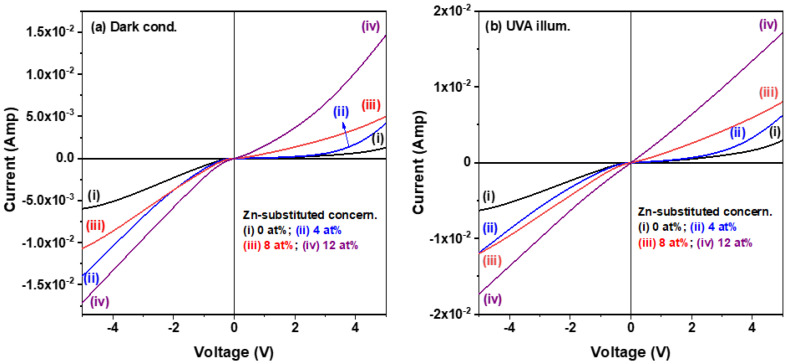
Comparison of current−voltage (I–V) characteristic curves of the CuI-based MSM photodetectors: (**a**) in a dark environment and (**b**) under UVA light illumination.

**Figure 7 materials-15-08145-f007:**
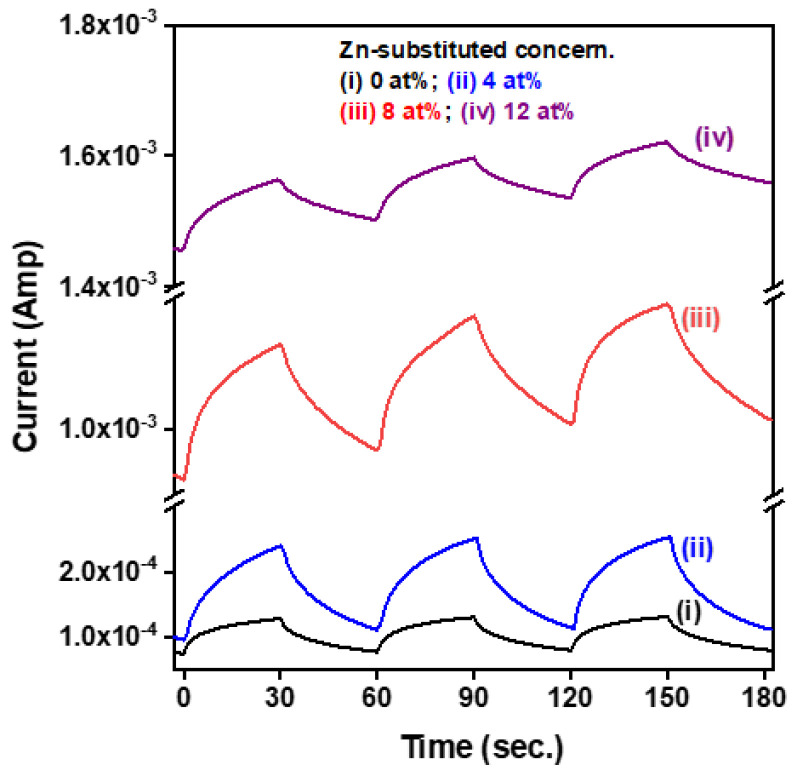
Time-resolved photoresponse characteristics of CuI-based MSM photodetectors under UVA light illumination and with an applied a bias voltage of 0.5 V. The interval time of periodically switching the UVA light on and off was 30 s.

**Table 1 materials-15-08145-t001:** Comparison of microstructural, optical and electrical characteristics of solution-processed Zn-substituted CuI thin films.

Concern. of Zn Substituting (at%)	Relative Intensity of (I_I(111)_/I_CuI(111) + I(111)_)	Average Crystalline Size (nm)	Optical Bandgap (eV)	Urbach Energy (meV)	Carrier Concern. (cm^−3^)	Hall Mobility (cm^2^/Vs)	Electrical Resistivity (Ω·cm)
0	0.250	7.23	3.00	81.6	4.45 × 10^18^	2.86	4.86 × 10^−1^
4	0.237	7.89	3.00	77.1	−4.41 × 10^18^	7.93	1.79 × 10^−1^
8	0.184	8.83	3.01	76.5	−4.34 × 10^18^	8.45	1.71 × 10^−1^
12	0.096	10.57	3.01	72.8	−4.01 × 10^18^	9.44	1.65 × 10^−1^

**Table 2 materials-15-08145-t002:** Mathematical relation and calculated results of photodetection properties of pure CuI and Zn-substituted CuI MSM photodetectors.

Parameter	* Mathematical Relation	Zn Substituted Concentration (at%)			
		0	4	8	12
Response current (Amp)	ΔI = I*_light_* − I*_dark_*	5.28 × 10^−5^	1.39 × 10^−4^	2.05 × 10^−4^	9.33 × 10^−5^
Sensitivity (unit less)	*S* = I*_light_* − I*_dark_*/I*_dark_*	0.68	1.25	0.21	0.06
Responsivity (mA/Watt)	*R* = ΔI/PA_0_	186	488	722	328
Specific detectivity (Jones)	*D* = R × (A_0_/2eI*_dark_*)^1/2^	1.37 × 10^8^	3.01 × 10^8^	1.51 × 10^8^	5.50 × 10^7^
External Quantum Efficiency (%)	*EQE* = (hc/eλ) × R	62.2	164	242	110

* I*_light_*: the photocurrent under illumination; I*_dark_*: the current detected in the dark condition; ΔI: measured current difference between the UV illumination and dark conditions; P: the power density of the incident light; A_0_: the effective illumination area; h: Planck’s constant; c: the speed of light; e is the electron charge; λ: the incident light wavelength.

**Table 3 materials-15-08145-t003:** Comparison of optoelectronic performance of different solution-processed MSM UV photodetectors based on wide bandgap semiconductors.

Sensing Layer	Electrodes/ Substrate	Method/ Temp. (°C)	Sensitivity (Unit Less)	Responsivity (mA/Watt)	Detectivity (×10^9^ Jones)	Illumin. Wavelength	Reference
ZnO	In/quartz	sol-gel/450	4.6	0.51	4.57	350 nm	[8]
ZnO	Al/glass	sol-gel/500	0.19	3.25	0.03	UVA	[13]
ZTO	Al/glass	sol-gel/500	2.95	0.14	<0.01	UVA	[16]
CuI	Ni/SiO_2_-Si	spin-coating/175	403	65.2	16.6	Blue-LED	[29]
CuI	Ni/PI	spin-coating/175	264	45.6	6.56	Blue-LED	[18]
CuI:Zn	Ni/glass	spin-coating/150	0.21	722	0.15	UVA	This work

## Data Availability

Not applicable.

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
