# Peer review of "The Role of Zn Substitution in Improving the Electrical Properties of CuI Thin Films and Optoelectronic Performance of CuI MSM Photodetectors"

_materials, 2022, doi:10.3390/ma15228145_

Round 1

Reviewer 1 Report

The manuscript is devoted to CuI semiconductor thin films doped with Zn , produced by a low-temperature solution process. It was found that adding up to 12% Zn in CuI film improves the crystallinity, increases the average crystalline size, changes the conductivity from p-type for intrinsic CuI to n-type for doped material, increases the free carriers concentration and mobility, and hence lowers the resistivity. The metal-semiconductor-metal (MSM) photodetectors fabricated using doped material have substantially better photo response than intrinsic.

The subject is interesting from scientific and applied points of view. The manuscript is well organized, the experiment is described in detail, the discussion of obtained results is convincing, and the conclusions are well supported.

However, before my recommendation for publication, several items should be corrected.

The Urbach energy (Fig. 5) was estimated in the same range of energy as the optical gap (Tauc plot) which is not the correct procedure.

227 “The Urbach energy was determined by calculating the inverse of the slope in the linear region of the absorption curve [26 ],”

Reference 26 does not contain optical measurements at all.

248 “…was in the range of -4.0´1018 to -4.45 ´1018 cm-3 , and the..”

Type mistake “-“

Reviewer 2 Report

1) Please add a note, that the copper and indium quantification by XPS is affected by an overlayer of adventitious hydrocarbons. This is related to the different kinetic energies of Cu 2p and I 3d, Cu 2p electrons being slower than I 3d electrons. That means that Cu 2p electrons are more severely attenuated by the hydrocarbon overlayer than the I 3d electrons, leading thus to an apparent excess of I.

2) What is the origin of the small asymmetry (shoulder at lower binding energy) of the Cu 2p3/2 emission?

3) How big is the error in the determination of the Urbach energy and are the observed differences between samples significant in this respect?

4) CuI is known for limited stability in ambient conditions. Can the authors comment on the stability of their samples?

5) an the authors please discuss the deterioration of photoresponse at 12% Zn, compared to the lower Zn concentrations?

6) “Wide-scan XPS surface analysis confirmed the presence of the constituent elements 164 of Cu, In, and Zn in the as-prepared CuI-based thin films” – should be I instead of In.

Reviewer 3 Report

The manuscript is very interested, and I recommended to publish as is.

Author Response

The authors thank the reviewer for their positive comment.

Reviewer 4 Report

In this manuscript, the authors reported the influence of Zn substitution concentration (0−12 at%) on the structural, optical, and electrical characteristics of CuI thin films. metal-semiconductor-metal (MSM) UV photodetectors were fabricated on glass substrates by the spin-coating technique. The authors presented adequate information about the structure, morphology, and composition of the adsorbate. Also, the description and interpretations of data are well organized. Overall, the work is interesting but needs some improvement in order to be published in the journal of Materials.

  1. The article is poorly written and English should be polished by a native speaker. There are many mistakes in writing and multiple grammatical mistakes throughout the manuscript that should be corrected.
  2. In the materials and methods section, details of the company from which chemical reagents were purchased should be given.
  3. Add the exact working area of the active part of the samples.
  4. In the results and discussion, the authors write: "Wide-scan XPS surface analysis confirmed the presence of the constituent elements of Cu, In, and Zn in the as-prepared CuI-based thin films and XPS fine-scan analysis revealed that the intensity of Zn 2p increased almost linearly with increases in Zn substitution concentration, indicating the successful incorporation of the expected amounts of impurity dopant of Zn into the CuI crystals (data not shown)." To clarify it, the XPS wide range survey spectra of all samples needs to be added.
  5. An appropriate discussion about the repeatability and durability of the as-prepared samples is missing.
  6. How are the performances here compared with state-of-the-art reports, especially MSM structure-based photodetectors? The readers would like to see a paragraph near the end of the manuscript before the conclusion dedicated to such a comparison. More recent literature is suggested to be included as a comparison.
  7. In all figures, "(a)" and "(b)" and "(c)" should be right or left inside or outside the figure with an identical format.

Round 2

Reviewer 1 Report

I have no further comments on the revised version.